# Controlled Hydrothermal Growth and Li^+^ Storage Performance of 1D VO_x_ Nanobelts with Variable Vanadium Valence

**DOI:** 10.3390/nano9040624

**Published:** 2019-04-17

**Authors:** Yuhan Jiang, Xiaowei Zhou, Xu Chen, Jia Wen, Linlin Guan, Mingxia Shi, Yang Ren, Zhu Liu

**Affiliations:** 1Department of Physics, School of Physics and Astronomy, Yunnan University, Kunming 650504, China; jiangyuhan1994@mail.ynu.edu.cn (Y.J.); chenxu@ynu.edu.cn (X.C.); wen__jia@163.com (J.W.); ninin0621@163.com (L.G.); smx303@126.com (M.S.); reny@lzu.edu.cn (Y.R.); 2Yunnan Key Laboratory of Micro/Nano-Materials and Technology, Yunnan University, Kunming 650504, China

**Keywords:** vanadium oxide nanobelts, controlled hydrothermal synthesis, variable vanadium valence, cathode materials, Li^+^ storage performance

## Abstract

One-dimensional (1D) vanadium oxide nanobelts (VO_x_ NBs) with variable V valence, which include V_3_O_7_·H_2_O NBs, VO_2_ (B) NBs and V_2_O_5_ NBs, were prepared by a simple hydrothermal treatment under a controllable reductive environment and a following calcination process. Electrochemical measurements showed that all these VO_x_ NBs can be adopted as promising cathode active materials for lithium ion batteries (LIBs). The Li^+^ storage mechanism, charge transfer property at the solid/electrolyte interface and Li^+^ diffusion characteristics for these as-synthesized 1D VO_x_ NBs were systematically analyzed and compared with each other. The results indicated that V_2_O_5_ NBs could deliver a relatively higher specific discharge capacity (213.3 mAh/g after 50 cycles at 100 mA/g) and median discharge voltage (~2.68–2.71 V vs. Li/Li^+^) during their working potential range when compared to other VO_x_ NBs. This is mainly due to the high V valence state and good crystallinity of V_2_O_5_ NBs, which are beneficial to the large Li^+^ insertion capacity and long-term cyclic stability. In addition, the as-prepared VO_2_ (B) NBs had only one predominant discharge plateau at the working potential window so that it can easily output a stable voltage and power in practical LIB applications. This work can provide useful references for the selection and easy synthesis of nanoscaled 1D vanadium-based cathode materials.

## 1. Introduction

The efficient storage and release of electric energy is an important topic in the field of energy storage research. At present, two kinds of energy storing devices, that is, lithium ion batteries (LIBs) and supercapacitors (SCs), have received great attention because of their respective features in high energy density and power density. As one of the typical energy storage units, LIBs are widely applied in various consumer electronics and electric tools since they have the advantages of large energy density, low self-discharge rate, no memory effect, etc. Electrode materials are directly responsible for the electrochemical performance of LIBs. It is crucial for the development of cathode materials with large capacities, high working voltage and high rate capability. Traditional cathode materials for LIBs, such as LiCoO_2_, LiMn_2_O_4_, LiNi_x_Co_y_Mn_z_O_2_ and LiFePO_4_, which only possess a specific capacity of ~120–180 mAh/g, could not meet the requirement of high-energy density for next generation LIBs [1]. Thus, it is necessary to explore new-type cathodes with large capacity. Among the diverse transition metal oxides, vanadium oxide (VO_x_) could act as a kind of promising cathode candidate with relatively high operating voltage and capacity during the Li^+^ intercalation/de-intercalation process, which was noted very early by researchers and battery manufacturers [2,3].

There are a wide variety of configurations for VO_x_ including V_2_O_5_, V_3_O_7_, VO_2_, V_6_O_13_, etc. To improve the electrochemical performance of the VO_x_ matrix (mainly the conductivity and structural stability upon the repeated charge/discharge), the commonly adopted methods are the design of nanostructure and further modifications, like doping, as well as combination with conductive frameworks [3,4,5]. The reported nanostructures involve 0-Dimensional (0D) (nanosphere and nanoparticle), 1D (nanobelt, nanofiber, nanorod and nanowire), 2D (nano-film, nanoplate and nanosheet) and 3D hierarchical (nanoflower, nanourchin and mesoporous structure) VO_x_, which were produced through various methods, such as template synthesis, self-assembly, hydrothermal process, sol-gel, spray pyrolysis, electro-spinning and chemical vapor deposition (CVD) [4,5,6,7,8,9]. The electrochemical Li^+^ storage performance of these VO_x_ nanostructures is significantly superior to those of their bulk counterparts and commercial cathodes. During a variety of VO_x_ nanostructures, one-dimensional (1D) VO_x_ has the advantages of convenient synthesis, structural simplicity and easy to scale-up production, which makes VO_x_ prospective as alternative cathode materials for LIBs. In recent years, the solvothermal method was mainly applied to prepare 1D V_3_O_7_, VO_2_ and V_2_O_5_ nanobelts, whose specific capacity and Li^+^ storage properties were investigated under different working voltage windows [10,11,12,13,14,15]. However, it is rarely reported that a series of 1D nanostructured VO_x_, which possess variable components and crystal structures, were acquired in a similar synthetic environment just by altering reaction reagents and combining them with a simple post-treatment. 

During our study, we propose an easy route to prepare 1D VO_x_ nanostructures with different crystalline textures and vanadium valence states based on hydrothermal treatment. For one thing, 1D V_3_O_7_·H_2_O and VO_2_ (B) nanobelts (NBs) with relatively low V valence were obtained through the previous solation of V_2_O_5_ powder and introduction of reducing ingredients (ethanol and CNTs) under hydrothermal condition. For another, V_2_O_5_ NBs with high V valence were further achieved by simple post-sintering to VO_x_ NBs in air under the circumstance of maintaining the 1D nanobelt shape. The microscopic morphologies, crystal lattices and V-O bonding of these 1D VO_x_ NBs were characterized and analyzed. When these VO_x_ NBs were used as active cathodes for LIBs, their Li^+^ storage properties were evaluated and compared systematically. According to this research, we get a simple method to obtain 1D VO_x_ NBs with variable V valence on a large scale. In addition, the micro-architecture and electrochemical Li^+^ insertion/extraction performance of these VO_x_ NBs were discussed and perceived.

## 2. Experimental Section

### 2.1. Sample Preparations

All reaction reagents were analytically pure grade and used as received.

Firstly, 1.82 g of V_2_O_5_ (Shanghai Kefeng industrial Co., Ltd., Shanghai, China) powder (referring to ESI, Appendix A) was dispersed into 75 ml of de-ionized water (self-produced by EPED-10TH apparatus from Nanjing Yipu Yida technology development Co. Ltd., Nanjing, China) with strong magnetic stirring, which formed a yellow suspension. Then, 5 ml of H_2_O_2_ (30%, mass concentration, Tianjin Fengchuan chemical reagent co. Ltd., Tianjin, China) was poured to the above V_2_O_5_ suspension under continuous agitation for 5 hours, generating translucent and orange V_2_O_5_ sol. After that, ~80 ml of V_2_O_5_ sol was further mixed with 0.2 ml of absolute ethanol (Tianjin Fengchuan) and transferred into a 100 ml of Teflon-lined hydrothermal autoclave (100 ml, Shanghai Lingtuo instrument and equipment Co., Ltd., Shanghai, China). The stainless steel reactor was sealed, kept at 180 °C for 2 days in an electric heating oven and cooled to room temperature naturally. The resultant black precipitate was rinsed, filtrated and dried at 80 °C. The sample obtained was labeled as V_3_O_7_·H_2_O nanobelts (NBs).

In order to study the effect of ethanol content on the final product, 2 ml of absolute ethanol was added to previously described V_2_O_5_ sol (80ml) and followed by hydrothermal treatment with identical operating procedures. The product was denoted as V_3_O_7_·H_2_O/VO_2_ (B)* NBs.

For further comparison, 2 ml of ethanol and 0.2 g of multi-walled carbon nanotubes (MWCNTs, Shenzhen Nanotech Port Co., Ltd., Shenzhen, China) were simultaneously introduced into the above mentioned V_2_O_5_ (80 ml) sol under ultrasonic oscillation, forming black turbid liquid that was also subjected to the same hydrothermal process. The MWCNTs used here were previously treated with the mixed concentrated acid (H_2_SO_4_:HNO_3_ = 3:1, volume ratio, Chengdu Kelong Chemical Reagent Factory, Chengdu, China) in order to create the surface functional groups and enhance their dispersibility in water. The collected powder has a dark black-blue color, which is labeled as VO_2_ (B) NBs/MWCNTs. The filling degree for autoclave during all hydrothermal reactions was controlled at 80%.

V_2_O_5_ NBs were prepared by directly sintering the previously acquired V_3_O_7_·H_2_O NBs in a tube furnace (OTF-1200X, Hefei Kejing-MTI technology Co., Ltd., Hefei, China) at 400 °C for 3 hours under air condition with the ramping rate of 5 °C/min.

### 2.2. Characterizations

A scanning electron microscope (SEM, QUANTA 200, FEI Company, Hillsboro, OR, USA) was employed to observe the surface morphologies of the as-prepared samples. The crystal structures of the products were determined by X-Ray Diffractometer (XRD, TTRIII-18KW, Rigaku, Tokyo, Japan) at the 2θ angle range of 10–90° (recording interval, 0.02°) with Cu Kα radiation (λ = 1.5406 Å). Transmission electron microscope (TEM, JEM-2100, JEOL, Akishima, Tokyo, Japan) equipped with the module for Selected area electron diffraction (SAED) was applied to examine the microstructures of samples. Material composition and atomic bonding for the products obtained were analyzed through Fourier transform infrared (FTIR) spectroscope with a Bruker-TENSOR27 (Bruker Optics, Karlsruhe, Germany) spectrometer at wave number region of 400–4000 cm^−1^ using KBr pellets. 

### 2.3. Electrochemical Tests

The cathode slurry was composed of active material (V_3_O_7_·H_2_O NBs, V_3_O_7_·H_2_O/VO_2_ (B)* NBs, VO_2_ (B) NBs/MWCNTs or V_2_O_5_ NBs), conductive acetylene black (Shenzhen Kejing Zhida-MTI technology Co., Ltd., Shenzhen, China) and polyvinylidene fluoride (PVDF, Shenzhen Kejing), with a mass ratio of 7:2:1, using N-methyl-2-pyrrolidone (NMP, Shenzhen Kejing) as an organic solvent. The slurry was uniformly coated onto the Al foil (Shenzhen Kejing) by a doctor blade and dried at 120 °C over 8 h in a vacuum to wipe off the NMP completely. Then, the formative cathode plate was cut into circular discs of 12 mm in diameter for electrochemical performance evaluations and the average active material loaded for all electrodes was ~1.4 mg/cm^2^. Coin cells (CR-2025, Shenzhen Kejing) were assembled by using the cathode disc as a working electrode, a porous polyolefin tri-layer membrane (Celgard 2325, Shenzhen Kejing) as a separator and Li metal as reference/counter electrode. A 1 M LiFP_6_ dissolved in ethylene carbonate/dimethyl carbonate/ethyl methylcarbonate (EC/DMC/EMC, 1/1/1, volume ratio, Shenzhen Kejing) was adopted as an electrolyte. The battery assembly was performed in a glove box with inert Ar atmosphere where both H_2_O and O_2_ gas were below 1 ppm. 

Cyclic voltammograms (CV) were recorded by an electrochemical workstation (CHI660, Shanghai Chenhua metrologic instruments Co., Ltd., Shanghai, China) at a scanning speed of 0.2 mV s^−1^ within the voltage region of 1.5–4 V or 2–4 V. Galvanostatic charge/discharge (GCD) tests were conducted using LAND (CT2001A, Wuhan Landian electronic Co., Ltd., Wuhan, China) cell-testing system between 1.5–4 V or 2–4 V under specific current densities. Electrochemical impedance spectroscopy (EIS) was also collected by CHI660 electrochemical workstation at the frequency region from 100 k to 0.01 Hz under the state of charge (SOC) of ~3 V with an alternating current (AC) signal of 5 mV. Nyquist plots were fitted by Z-view software.

## 3. Result and Discussion

Figure 1 shows the synthetic process for V_3_O_7_·H_2_O NBs, VO_2_ (B) NBs/MWCNTs and V_2_O_5_ NBs, whose microscopic molecular crystal structures are illustrated in the insets. The V_3_O_7_·H_2_O NBs with V^4+^ and V^5+^ coexisting was prepared through hydrothermal reaction using V_2_O_5_ sol as a precursor and ethanol as a weak reducing agent. When increasing the amount of ethanol and simultaneously introducing MWCNTs as an auxiliary reducing ingredient, VO_2_ (B) NBs with predominant V^4+^ were obtained at the same condition. After direct sintering in the air at 400 °C for 3 h, V_3_O_7_·H_2_O NBs were transformed into V_2_O_5_ NBs with complete V^5+^.

XRD patterns of all nanostructured VO_x_ samples prepared under different conditions are presented in Figure 2. Through the crystal structure analysis, the pattern of Figure 2a could be ascribed to the orthorhombic phase of V_3_O_7_·H_2_O (PDF# 28-1433) with characteristic peaks of (020), (120), (130), etc. Two sets of diffraction peaks can be deconstructed from Figure 2b, which belongs to the orthorhombic V_3_O_7_·H_2_O (major component) and monoclinic VO_2_ (B) (minor component with asterisk mark, PDF# 31-1438) with peaks of (200), (110), (−311), etc., respectively. This indicates that the ethanol, as a weak reducing agent, would lead to the conversion of V^5+^ to V^4+^ and meanwhile be oxidized to aldehyde itself. When maintaining a relatively high ethanol content, like the sample V_3_O_7_·H_2_O/VO_2_ (B)* NBs, and bringing in MWCNTs, the monoclinic VO_2_ (B) was primarily acquired as shown in Figure 2c. This result shows that MWCNTs would play the role of inductive reducibility during hydrothermal treatment, which may owe to the oxygen-contained functional groups on the surface of MWCNTs generated during the mixed acid treatment and hydrothermal reaction. There is a broad peak envelope at 2θ ≈ 26° in Figure 2c, which was caused by the (002) plane of amorphous MWCNTs during the sample VO_2_ (B) NBs/MWCNTs [16,17]. For sample V_2_O_5_ NBs, whose XRD pattern in Figure 2d confirms its orthorhombic V_2_O_5_ phase (PDF# 41-1426), with a typical crystal plane of (200), (001), (101), etc., illustrating the transformation of V^4+^ to V^5+^ when V_3_O_7_·H_2_O was subject to sintering in air.

SEM images of V_3_O_7_·H_2_O NBs (a), V_3_O_7_·H_2_O/VO_2_ (B)* NBs (b), VO_2_ (B) NBs/MWCNTs (c) and V_2_O_5_ NBs (d) are shown in Figure 3. It can be seen that all of them display one-dimensional nanobelted morphology with a width of 100–300 nm and a length of over 10 microns. Due to the role of energy modulation under hydrothermal conditions, the growth kinetics of the vanadium oxide precursor is faster in a specific direction during the process of self-assembly crystallization and, conversely, the growth in other orientations are inhibited, resulting in a one-dimensional VO_x_ nano-belted morphology. In Figure 3c, VO_2_ (B) NBs were evenly mingled with MWCNTs. In addition, SEM images of VO_2_ (B) NBs/MWCNTs (Figure 3c) and V_2_O_5_ NBs (Figure 3d) also demonstrate that the adding of MWCNTs or post-sintering will not change the 1D nanostructure of VO_x_ samples, but result in the further decrease or increase of V valence as recognized by the XRD detections.

Figure 4 illustrates TEM and HRTEM (High-Resolution TEM) images of V_3_O_7_·H_2_O NBs (a,b), VO_2_ (B) NBs/MWCNTs (c,d) and V_2_O_5_ NBs (e,f). We can see from the TEM images (a,c,e) that the diameter of these samples distributes approximately between 100–300 nm, which is in accordance with previous SEM observations. Figure 4b gives the lattice fringes of (101) and (200) planes with distances of 0.338 nm and 0.466 nm for V_3_O_7_·H_2_O NBs [18,19]. The inset shows the corresponding SAED pattern. The HRTEM image of VO_2_ (B), whose (−201) crystal plane is marked with the space of ~0.5 nm, is presented in Figure 4d. Figure 4f demonstrates the (110) crystal plane for V_2_O_5_ NBs with a distance of 0.341 nm, as well as its SAED pattern in the upper right corner [20,21]. Thus, it could be learned that the results of TEM characterizations are consistent with those values from the calculations of the associated XRD peak positions based on the Scherrer formula.

FTIR spectra of the as-prepared four samples are shown in Figure 5. The characteristic absorption peaks near 1000 cm^−1^ (976, 980, 1007, 1016, 1020 and 1022 cm^−1^) can be attributed to stretching vibrations of terminal oxygen bonds (V=O). It should be noted that the appearance of relatively low wave numbers (976, 980 and 1007 cm^−1^) for V=O vibrations is caused by the V^4+^ in V_3_O_7_·H_2_O NBs, V_3_O_7_·H_2_O/VO_2_ (B)* NBs and VO_2_ (B) NBs/MWCNTs because the bond length of V^4+^=O is longer than that of V^5+^=O, leading to the decrease of frequency [22]. The peaks at 802 cm^−1^ and below 600 cm^−1^ (449, 480, 546, 557, 567 and 577 cm^−1^) emerge because of the vibration modes of the doubly coordinated oxygen (V-O-V) bonds in V_2_O_5_ and the triply coordinated oxygen (V_3_–O) bonds within VO_x_, respectively [23]. The slight shifting for these V–O coordinated vibration peaks among different samples may be connected with the microscopic stress change in the crystal lattice. Besides, the absorption peaks at 1628–1633 cm^−1^ and 3412–3444 cm^−1^ could be ascribed to the bending and stretching vibrations of H–O bonds from the absorbed/bound water in samples, respectively [24,25]. The wave-number at ~2350 cm^−1^ is due to the contamination of CO_2_ molecules in the measuring atmosphere [26].

The initial three CV curves of the four samples between 1.5–4 V are depicted in Figure 6. 

For V_3_O_7_·H_2_O NBs, four pairs of stable cathodic/anodic peaks appear at ~3.30/3.37, 3.09/3.21, 2.56/2.92 and 1.80/2.11 V, in which two pairs of redox peaks at ~3.30/3.37 and 3.09/3.21 V between 3.0–3.5 V correspond to the reaction: V_3_O_7_·H_2_O + Li^+^ + e^−^ ↔ LiV_3_O_7_·H_2_O.(1)

The other pair of peaks at ~2.56/2.92 V between 2.3–3.0 V can be ascribable to the reaction: LiV_3_O_7_·H_2_O + Li^+^ + e^−^ ↔ Li_2_V_3_O_7_·H_2_O.(2)

Another pair of peaks at ~1.80/2.11 V in the region of 1.5–2.3 V is related with the reaction: Li_2_V_3_O_7_·H_2_O + xLi^+^ + xe^−^ ↔ Li_2+x_V_3_O_7_·H_2_O (x < 1).(3)

Both Equations (1) and (2) involve the conversion of V^5+^↔V^4+^, whereas Equation (3) refers to the partial V^4+^↔V^3+^ [27,28]. The CV curves of VO_2_ (B) NBs/MWCNTs are shown in Figure 6c. One pair of predominant redox peaks emerges at ~2.48/2.78 V, which is the typical trait for the monoclinic VO_2_ (B) phase during the repeated Li^+^ insertion/extraction, corresponding to the reaction [29]:VO_2_ (B) + xLi^+^ + xe^−^ ↔ Li_x_VO_2_ (B) (x ≈ 0.5)(4)

According to the analysis of a previous XRD, we know that V_3_O_7_·H_2_O/VO_2_ (B)* NBs contain a small quantity of the VO_2_ (B) phase. Thus, the overall outline of the CV curves for V_3_O_7_·H_2_O/VO_2_ (B)* NBs in Figure 6b is similar to that of the V_3_O_7_·H_2_O NBs, but there is an upward anodic shoulder peak appearing at ~2.76 V and a slightly lower shift for the downward cathodic peak from 2.56 to 2.52 V. Further, the relative peak area between 2.3–3.0 V for V_3_O_7_·H_2_O/VO_2_ (B)* NBs is larger compared to the V_3_O_7_·H_2_O NBs. These changes are concerned with the redox process of V^4+^↔V^3+^ in the VO_2_ (B) phase contained. The solid and dotted lines in Figure 6d present the CV curves of the V_2_O_5_ NBs at the voltage sweep range of 1.5–4 V and 2–4 V, respectively. During the initial Li^+^ intercalation between 1.5–4 V, the cathodic peaks at ~3.33, 3.08 and 2.14 V correspond to the phase transitions of α-Li_x_V_2_O_5_ to ε-Li_x_V_2_O_5_, ε-Li_x_V_2_O_5_ to δ-Li_x_V_2_O_5_ (x < 1) and δ-Li_x_V_2_O_5_ to γ-Li_x_V_2_O_5_ (1 < x < 2), respectively. When the working voltage lowers below 2 V, the irreversible phase transition happens from γ-Li_x_V_2_O_5_ to ω-Li_x_V_2_O_5_(x > 2) [30,31]. Therefore, the anodic peaks associated with the phase transitions above do not arise in the subsequent Li^+^ extraction process. There is only a wide anodic peak at ~2.71 during the following 2nd–3rd cycles, illustrating the amorphous features and irreversible structural change of V_2_O_5_ NBs operating in this voltage region (1.5–4 V). When V_2_O_5_ NBs work at 2–4 V, three pairs of primary redox peaks appear at ~3.33/3.45, 3.08/3.35 and 2.14/2.56 V, indicating the reversible phase transition between α-Li_x_V_2_O_5_ ↔ γ-Li_x_V_2_O_5_ (x < 2) [31,32]. Thus, the subsequent GCD evaluation for V_2_O_5_ NBs was performed between 2–4 V.

GCD profiles of four samples are plotted in Figure 7 at the 1st, 2nd, 5th, 20th and 50th charge/discharge cycles under 100 mA/g. Because VO_x_ NBs non-embedded by Li^+^ are at a full charge state in the beginning, there is only the 1st discharge process for all samples. With regard to the V_3_O_7_·H_2_O NBs in Figure 7a, four voltage plateaus emerged in sequence at ~3.3, 3.1, 2.6 and 1.8 V during the discharge (Li^+^ intercalation), which are related with the four downward cathodic peaks in its CV curve. Conversely, the voltage plateaus turn up at ~2.1, 2.8, 3.2 and 3.4 V sequentially during the charge (Li^+^ de-intercalation), corresponding to its upward anodic peaks in the CV. For the VO_2_ (B) NBs/MWCNTs in Figure 7c, there was an obvious discharge and charge plateau at 2.5 V and 2.6 V, respectively, which are concerned with the pair of dominant redox peaks in its CV. The GCD profile of the V_3_O_7_·H_2_O/VO_2_ (B)* NBs in Figure 7b is similar to that of V_3_O_7_·H_2_O NBs. However, the discharge plateau at ~2.5 V was elongated visibly. Further, the charge plateau at ~2.8 V was also lengthened and split into two close plateaus. These variations in GCD for the V_3_O_7_·H_2_O/VO_2_ (B)* NBs are consistent with the changes in its CV curves. As for the V_2_O_5_ NBs in Figure 7, when working between 2–4 V, the stable GCD profile was formed through the first few discharge/charge activations, whose reversible plateaus correspond to the prime redox peaks in its CV (dotted line). From the standpoint of median discharge voltage (MDV), V_3_O_7_·H_2_O NBs, VO_2_ (B) NBs/MWCNTs and V_2_O_5_ NBs show the MDV values of ~2.61–2.70 V, ~2.50 V and ~2.68–2.71 V during their operating voltage ranges, respectively.

The cycling performances of the four samples at 100 mA/g are successively displayed in Figure 8a–d. V_3_O_7_·H_2_O NBs in Figure 8a delivered a highest specific capacity of 219.8 mAh/g at the first discharge and kept at 181.9 mAh/g after 50 cycles. As shown in Figure 8b, V_3_O_7_·H_2_O/VO_2_ (B)* NBs presented the highest discharge capacity of 222.2 mAh/g for the first cycle and remained at 176.2 mAh/g on the 50th discharge. The VO_2_ (B) NBs/MWCNTs in Figure 8c gave the largest value of 178.8 mAh/g at the first discharge and maintained at 148.9 mAh/g for the 50th cycle. Although the specific capacity of VO_2_ (B) NBs is unsatisfactory, its charge/discharge plateau is very flat. During practical usage, VO_2_ (B) NBs will output stable voltage and power. The highest capacity of 240.4 mAh/g was released at the third discharge for V_2_O_5_ NBs after initial two activation cycles in Figure 8d. When the cycling number went to the 50th cycle, this value remained at 213.3 mAh/g. We can see from the above cycling diagrams that the capacities for all samples exhibit periodic fluctuation with cyclic numbers, which are due to the large diurnal temperature variation between 10–21 °C in the local lab. This result shows that temperature has a non-negligible effect on the Li^+^ storage property for all VO_x_ samples. It is noteworthy that V_2_O_5_ NBs possesses higher capacity and cycling stability owing to their complete V^5+^ and better crystallinity. Thus, the rate capability of V_2_O_5_ NBs was further evaluated, as illustrated in Figure 8e. The typical CD curves at the last cycle under different current densities of 50, 100, 200, 400 and back to 50 mA/g are shown in Figure 8f, whose corresponding discharge capacities were 237.1, 217.1, 199.9, 184.9 and 221.8 mAh/g, respectively.

Nyquist plots and Warburg coefficients (A_w_) linear fittings of four samples at the SOC of ~3 V are illustrated in Figure 9a,b, respectively. Nyquist plots are composed of a compressed semicircle arc at a high-frequency region and an oblique straight line at low-frequency region, in which the X-axis intercept of semicircle on the extreme left denotes the equivalent series resistance (R_e_), the span of the semicircle is associated with the charge transfer resistance (R_ct_) at the electrolyte/electrode interface and the sloping line is connected with the Li^+^ diffusion in the solid electrode [33]. Based on the equivalent circuit (in which CPE is the constant-phase element, Wo represents the Warburg impedance) in the inset of Figure 9a, we can get the fitted values of R_e_ and R_ct_ for four samples, as listed in Table 1. All of their R_e_ values are small and distribute at 4–6 Ω. The R_ct_ of VO_2_ (B) NBs/MWCNTs is the lowest, which could be due to incorporation of conducting MWCNTs as well as the existence of predominant V^4+^ in the sample [34,35]. The lower R_ct_ is beneficial to the migration and transport of Li^+^ at the electrolyte/electrode interface.

To discuss the diffusion of Li^+^ during the active VO_x_ NBs, we need to access the Li^+^ diffusion coefficient (D_Li+_) of the samples, which can be expressed as [36]: (5)$$D_{{\mathrm{Li}}^{+}}=\frac{R^{2}T^{2}}{2n^{4}F^{4}A^{2}C^{2}A_{w}{}^{2}}$$
where, R (ideal gas constant), T (absolute temperature), n (electron transfer number per mole), F (Faraday’s constant), A (geometric area of electrode) and C (molar concentration of Li^+^) are fixed experimental parameters under certain sample and test condition. A_w_ is the Warburg coefficient that is related to the Li^+^ diffusion process. Because D_Li+_ ∝ A_w_^−2^, we can use the value of A_w_ to quantitatively evaluate the Li^+^ diffusion. According to the equation Z′ = R_e_ + R_ct_ + A_w_ω^−1/2^ (6) at the low-frequency region, a linear relationship between Z′ and ω^−1/2^ can be observed for all samples, as shown in Figure 9b. The slope of a straight line is the value of A_w_ as listed in Table 1, which would be acquired by linear fitting. The results show that V_3_O_7_·H_2_O/VO_2_ (B)* NBs and V_2_O_5_ NBs exhibit a relatively lower A_w_ at the given SOC, signifying the preferable diffusion behavior.

## 4. Conclusions

In this work, three kinds of 1D VO_x_ NBs, referring to V_3_O_7_·H_2_O NBs, VO_2_ (B) NBs and V_2_O_5_ NBs, which have different crystalline structure and vanadium valence, were synthesized under hydrothermal conditions merely by adjusting and adding the reductive ingredients (ethanol and MWCNTs), as well as conducting post-sintering treatment. XRD and SEM demonstrated the crystal structures and morphologies of acquired VO_x_ NBs (showing a typical width of 100–300 nm and over 10 microns in length). TEM and FTIR revealed their micro-architectures and components. The electrochemical Li^+^ storage properties of these VO_x_ NBs were systematically analyzed and compared with each other as cathode materials for LIBs. Among them, V_2_O_5_ NBs exhibited a relatively larger capacity, higher median discharge voltage and more stable cycling performance, which can be attributed to its higher V valence state and better crystallization. From the perspective of actual use, although VO_2_ (B) NBs showed an unsatisfactory specific capacity (148.9 mAh/g after 50 cycles at 100 mA/g), it displayed just one flat discharge platform at the testing potential window, which benefits the stable output of working voltage and power in practical battery applications. These representative 1D VO_x_ NBs derived from a simple hydrothermal reaction could be used as promising cathode candidates for future LIBs according to actual demands. 

## Figures and Tables

**Figure 1 nanomaterials-09-00624-f001:**
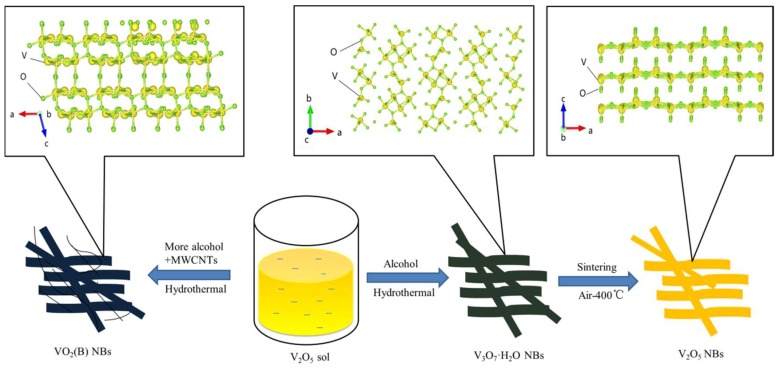
Schematic diagram for preparation of V_3_O_7_·H_2_O NBs, VO_2_ (B) NBs/MWCNTs and V_2_O_5_ NBs as well as their corresponding molecular crystal structures.

**Figure 2 nanomaterials-09-00624-f002:**
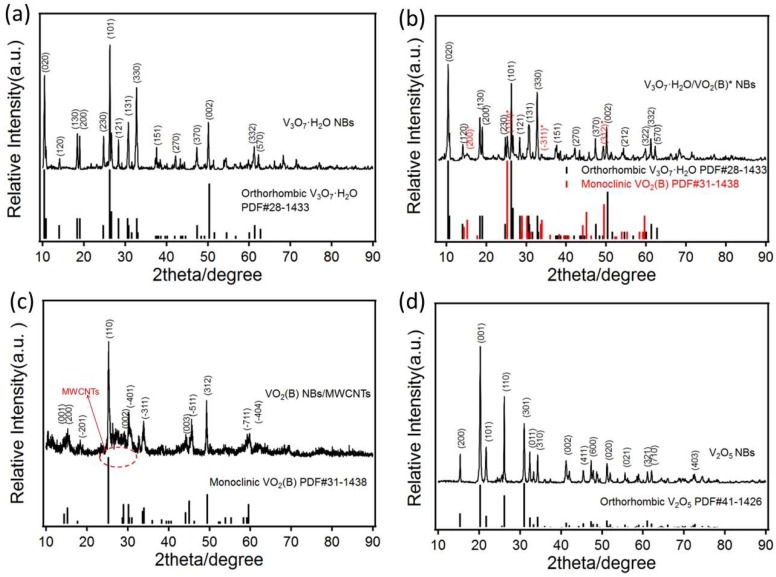
XRD patterns of V_3_O_7_·H_2_O NBs (**a**), V_3_O_7_·H_2_O/VO_2_ (B)* NBs (**b**), VO_2_ (B) NBs/MWCNTs (**c**) and V_2_O_5_ NBs (**d**).

**Figure 3 nanomaterials-09-00624-f003:**
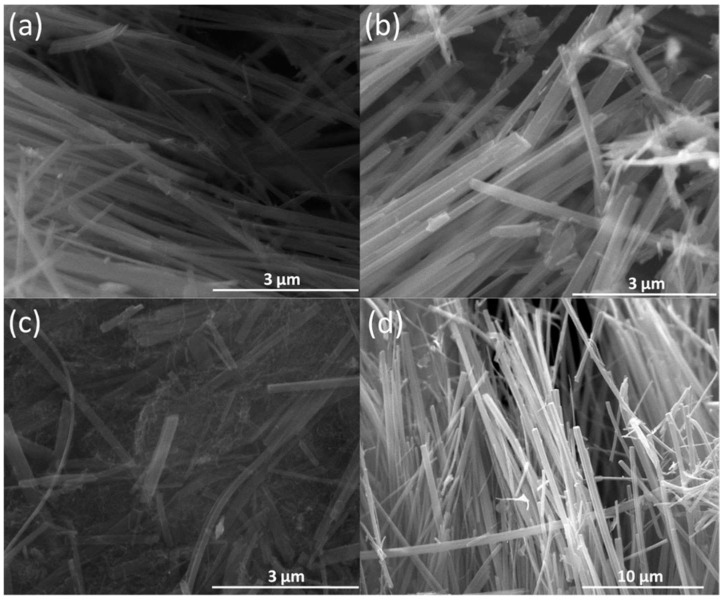
SEM images of V_3_O_7_·H_2_O NBs (**a**), V_3_O_7_·H_2_O/VO_2_ (B)* NBs (**b**), VO_2_ (B) NBs/MWCNTs (**c**) and V_2_O_5_ NBs (**d**).

**Figure 4 nanomaterials-09-00624-f004:**
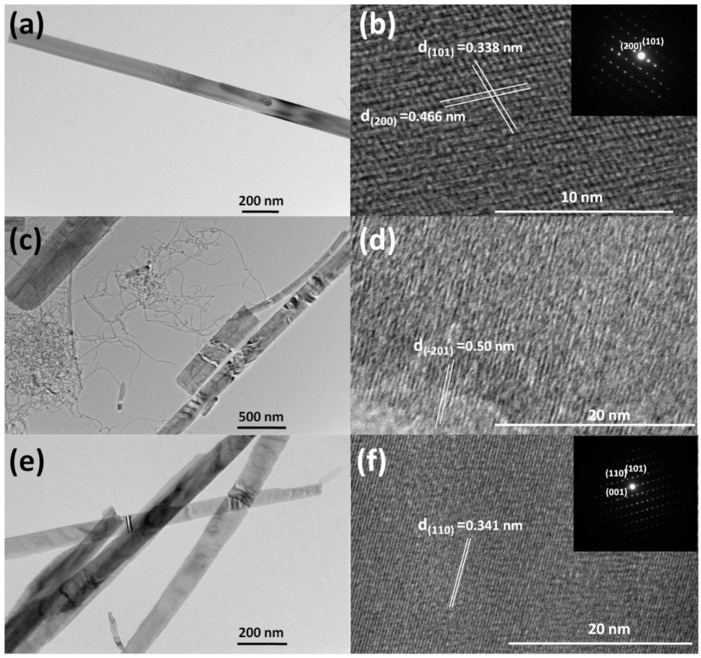
TEM and HRTEM images of V_3_O_7_·H_2_O NBs (**a**,**b**), VO_2_ (B) NBs/MWCNTs (**c**,**d**) and V_2_O_5_ NBs (**e**,**f**); the insets in (**b**) and (**f**) show SAED patterns.

**Figure 5 nanomaterials-09-00624-f005:**
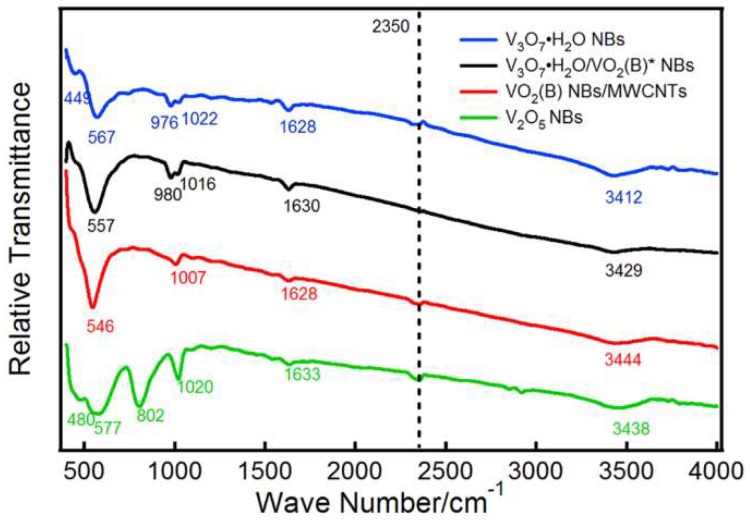
FTIR spectra of V_3_O_7_·H_2_O NBs, V_3_O_7_·H_2_O/VO_2_ (B)* NBs, VO_2_ (B) NBs/MWCNTs and V_2_O_5_ NBs between 400–4000 cm^−1^.

**Figure 6 nanomaterials-09-00624-f006:**
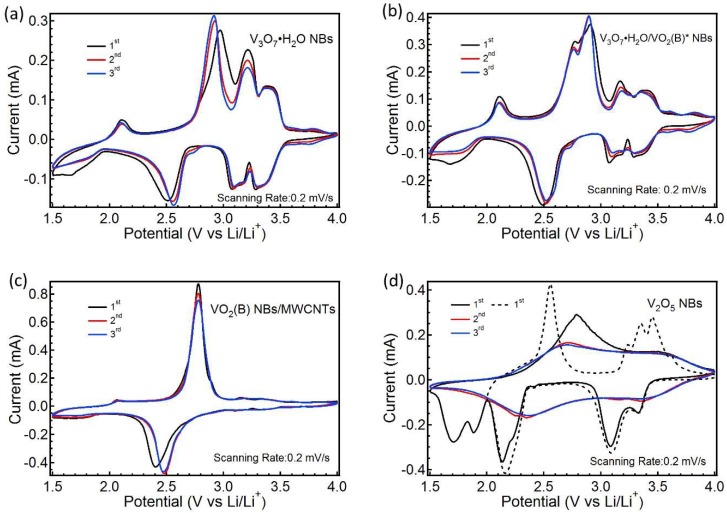
CV curves of V_3_O_7_·H_2_O NBs (**a**), V_3_O_7_·H_2_O/VO_2_ (B)* NBs (**b**), VO_2_ (B) NBs/MWCNTs (**c**) and V_2_O_5_ NBs (**d**) with a scanning speed of 0.2 mV s^−1^.

**Figure 7 nanomaterials-09-00624-f007:**
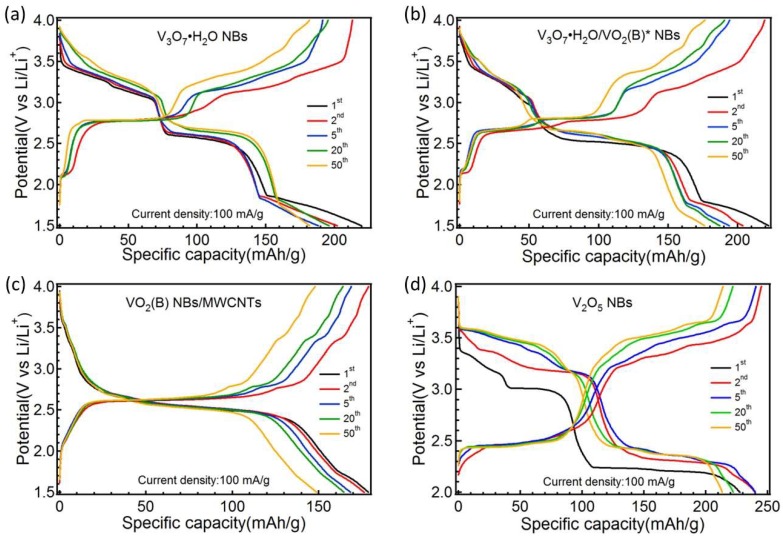
Galvanostatic charge/discharge profiles of V_3_O_7_·H_2_O NBs (**a**), V_3_O_7_·H_2_O/VO_2_ (B)* NBs (**b**), VO_2_ (B) NBs/MWCNTs (**c**) and V_2_O_5_ NBs (**d**) under different cycles at the current density of 100 mA g^−1^.

**Figure 8 nanomaterials-09-00624-f008:**
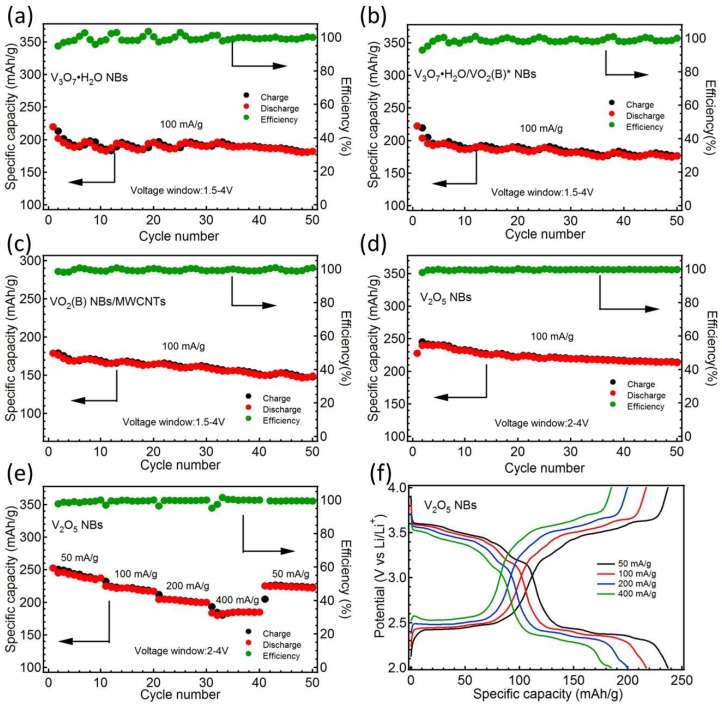
Cycling performances and coulombic efficiencies of V_3_O_7_·H_2_O NBs (**a**), V_3_O_7_·H_2_O/VO_2_ (B)* NBs (**b**), VO_2_ (B) NBs/MWCNTs (**c**) and V_2_O_5_ NBs (**d**) at 100 mA g^−1^; rate capability (**e**) and typical galvanostatic charge/discharge (GCD) profiles (**f**) of V_2_O_5_ NBs under varying current densities.

**Figure 9 nanomaterials-09-00624-f009:**
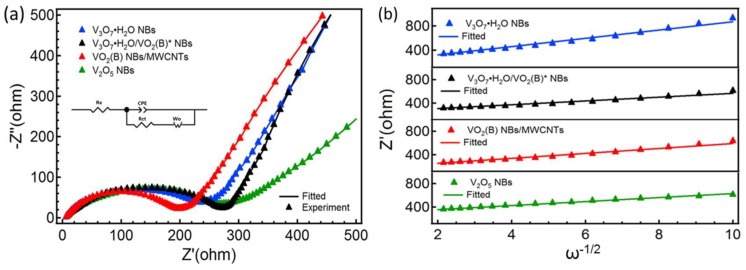
Nyquist plots (**a**) and dependences (**b**) of real resistance (Z′) on reciprocal of square root of angular frequency (ω) for all samples; the inset in (**a**) illustrating the equivalent circuit.

**Table 1 nanomaterials-09-00624-t001:** Fitted values of R_e_, R_ct_ and A_w_ for all prepared samples.

Samples	R_e_ (Ω)	R_ct_ (Ω)	A_w_ (Ω·s^−1/2^)
V_3_O_7_·H_2_O NBs	5.1	210.5	69.0
V_3_O_7_·H_2_O/VO_2_(B)* NBs	4.6	276.9	33.1
VO_2_(B) NBs/MWCNTs	5.1	199.8	42.9
V_2_O_5_ NBs	5.7	230.1	34.4

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
