# Peer review of "Controlled Hydrothermal Growth and Li+ Storage Performance of 1D VOx Nanobelts with Variable Vanadium Valence"

_nanomaterials, 2019, doi:10.3390/nano9040624_

Round 1
Reviewer 1 Report
The manuscript deals with the synthesis, structural characterization and electrochemical characterization of vanadium oxide nanostructures (nano belts). In detail, the authors present an adequate review of literature with focus on vanadium nanomaterials and they provide a clear description of their experimental procedures. The prepared vanadium oxide nano structures were intensively characterized by XRD, SEM, TEM, HRTEM and FT-IR. The results shown are clearly presented and discussed with reference to other data from recent literature. The major part of the manuscript deals with the very detailed electrochemical characterization of the prepared vanadium oxide (CV, charge/discharge curves, cycling experiments and impedance spectroscopy). This section covers all aspects from starting from the redox behavior until cycling stability and the assessment of the Li+ diffusion coefficient. Again, the results are clearly presented in the text as well as in the figures, a comprehensive discussion is made which goes far beyond the studies made in other publications. The manuscript is written in an excellent style.
I suggest to publish the manuscript as it is.
Author Response
Dear reviewer:
It is grateful for your careful review and objective evaluation to our work.
We have further modified our work and provided relevant electronic supplementary information (ESI, supporting information) to illustrate the points given in the article.
Please see the revised manuscript and electronic supplementary information.
Best wishes,
Yuhan Jiang

Reviewer 2 Report
This manuscript demonstrated controlled hydrothermal growth of VOx using different reaction conditions, and tested them as a cathode for lithium ion batteries. I think this manuscript can be published after addressing the following issues.
1. What is the morphology of the precursor V2O5? The authors used V2O5 as a starting materials to transform them into 1D VOx using reducing agents. The morphology of original V2O5 should be shown.
2. The authors used EtOH as a reducing agent. It should be clearly explained how EtOH can act as a reducing agent to produce VOx.
3. MWNT was also used as a reducing agent. Pristine MWNT can be dispersed in water? Or surface functionalization was performed? Can it be used as a reducing agent?
4. The authors should explain how and why morphological change of VOx takes place after the treatment
Author Response
Dear reviewer:
Thanks for your patient review and helpful comments.
The following content is the answers to your questions and queries:
1. What is the morphology of the precursor V2O5? The authors used V2O5 as a starting materials to transform them into 1D VOx using reducing agents. The morphology of original V2O5 should be shown.
SEM images of commercial V2O5 powders and the corresponding description are supplemented in the supporting information (electronic supplementary information, ESI). The size distribution of V2O5 particles is not uniform (the diameter ranging from hundreds of nanometers to a dozen of micrometers) and their shape is also irregular.
2. The authors used EtOH as a reducing agent. It should be clearly explained how EtOH can act as a reducing agent to produce VOx.
Actually, the hydrothermal reaction itself could provide a reductive atmosphere[1]. EtOH can behave as a kind of weak reducing agent in the hydrothermal environment. Some reports on the use of ethanol as a reducing agent have been reported [2, 3]. EtOH will be oxidized and converted to aldehydes and simultaneously vanadium element would be reduced to the lower valence state (for example: from V5+ to V4+, the possible reaction equation is provided in the supporting information).
3. MWNT was also used as a reducing agent. Pristine MWNT can be dispersed in water? Or surface functionalization was performed? Can it be used as a reducing agent?
MWCNTs are not well dispersed in water and it only shows the temporary state of suspension in solution. However, under the special high-temperature and pressure liquid environment of hydrothermal reaction, MWCNTs can be dispersed evenly to ensure their sufficient contact with vanadium oxide precursors. Because the MWCNTs here used were previously treated with mixed concentrated acids in order to get better dispersion in liquid condition. Generally, the mixed concentrated acids are H2SO4 (>70 wt%) and HNO3 (~65 wt%) according to 3:1 (volume ratio). The acid treatment process will create some surface hydroxyl, carbonyl and carboxyl functional groups [4, 5]. These functional groups on MWCNTs contribute to the further reduction of vanadium oxide precursors under the hydrothermal condition and the presence of EtOH.
4. The authors should explain how and why morphological change of VOx takes place after the treatment
V2O5 itself has low solubility in water. However, under the special high temperature and high pressure hydrothermal condition, the commercial V2O5 powder will disperse to form the metastable VOx precursor. Owing to the role of energy modulation under hydrothermal condition, the growth kinetics of vanadium oxide precursor is faster in a specific direction during the process of self-assembly crystallization and conversely the growth in other orientations are inhibited, resulting in a one-dimensional VOx nano-belted morphology. As some literatures reported [6, 7], we know that hydrothermal reaction is a common method that can be adopted to prepare a series of one-dimensional metal oxide nanomaterials.
In the post-sintering of VOx NBs in air, the V valence state will increase by in-situ oxidation, but the 1D morphology of VOx NBs could be maintained and form V2O5 NBs with high V valence state.
References:
1 Huang, H.H., et al., Structural Evolution of Hydrothermally Derived Reduced Graphene Oxide. Sci Rep, 2018. 8(1): p. 6849.
2. He, H., et al., Precipitable silver compound catalysts for the selective catalytic reduction of NOx by ethanol. Applied Catalysis A: General, 2010. 375(2): p. 258-264.
3. Sylvain Broussy, R.W.C., and David B. Berkowitz, Enantioselective, Ketoreductase-Based Entry into Pharmaceutical Building Blocks: Ethanol as Tunable Nicotinamide Reductant. ORGANIC LETTERS, 2009: p. 305-308.
4. Yue, L., et al., Highly hydroxylated carbon fibres as electrode materials of all-vanadium redox flow battery. Carbon, 2010. 48(11): p. 3079-3090.
5. Zhang, L., et al., Mild hydrothermal treatment to prepare highly dispersed multi-walled carbon nanotubes. Applied Surface Science, 2011. 257(6): p. 1845-1849.
6. Liu, H., Facile Preparation of 1D a-MnO2 as Anode Materials for Li-ion Batteries. International Journal of Electrochemical Science, 2016: p. 8964-8971.
7. Xing, X., et al., Synthesis of mixed Mn–Ce–Ox one dimensional nanostructures and their catalytic activity for CO oxidation. Ceramics International, 2015. 41(3): p. 4675-4682.
Best wishes,
Yuhan Jiang
